# The Domestication of Humans

**Robert G. Bednarik** 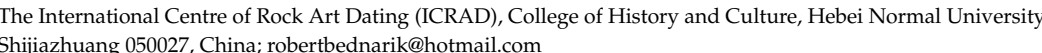

The International Centre of Rock Art Dating (ICRAD), College of History and Culture, Hebei Normal University, Shijiazhuang 050027, China; robertbednarik@hotmail.com

**Definition:** The domestication of humans is not an issue of domesticity but of the effects of the domestication syndrome on a hominin species and its genome. These effects are well expressed in the 'anatomically modern humans', in their physiology, behavior, genetic defects, neuropathology, and distinctive neoteny. The physiological differences between modern (gracile) humans and their ancestors, robust *Homo sapiens* types, are all accounted for by the domestication syndrome. From deductions we can draw about early human behavior, it appears that modifications are attributable to the same cause. The domestication hypothesis ascribes the initiation of the changes to selective breeding introduced by the consistent selection of neotenous features. That would trigger genetic pleiotropy, causing the changes that are observed.

**Keywords:** domestication syndrome; pleiotropy; human evolution; modern humans; neoteny; auto-domestication

## 1. Introduction

The origin of our subspecies, *Homo sapiens sapiens*, is arguably the most divisive topic in hominin evolution. It is generally agreed that during the Late Pleistocene, robust hominins, including the Neanderthals and Denisovans, were replaced with gracile forms called 'anatomically modern humans'. At least in Eurasia, this occurred over a relatively short period, a geological instant. A hypothesis was developed during the 1980s, proposing that our kin first arose in an unspecified part of Sub-Saharan Africa. From there, it colonized the three Old World continents and Australia. African Eve's progeny was not interfertile with the primitive humans they encountered, who were culturally, technologically, cognitively, and intellectually inferior to them. So, they either outcompeted or exterminated them in history's most comprehensive genocide. This 'replacement hypothesis' soon took over nearly the entire discipline, despite the lack of any archaeological, paleoanthropological, or genetic evidence in its favor. Its most severe shortcoming, however, was that it failed to explain the issue. What caused the changes differentiating gracile from robust Late Pleistocene hominins? Science is extensively based on the principle of causation. The causes and effects of the transition from robust to gracile hominins still need to be elucidated.

For instance, the replacement hypothesis does not explain what could have caused the change from the dysteleological progress of evolution to the apparent teleology of cultural development, or why we graciles are such neotenous primates, or what could have suspended the inherent laws of biological evolution. Nor does this failed hypothesis (refuted by the genetic demonstration that the robust and gracile humans were interfertile [1–8]) explain why natural selection failed to select against numerous deleterious genetic predispositions and defects. It also fails to elucidate why brain illness etiologies suggest that they involve mostly the same areas of the brain that are the phylogenetically latest; or why other extant primates are largely, if not entirely, free of such pathologies. Nor does it explain why the graciles are experiencing brain atrophy or any other of the many differences that set them apart from the preceding robust humans [9]. Until 2008, the preservation of the mutations involved in the significant deleterious etiologies remained essentially unexplained, leading to the proposal of a unified theory of human self-domestication [10]. It explains

not just all the questions posed here; it explicates the causes of all factors that constitute the human condition as we know it [11].

When applied to humans, the popular concept of domestication is related to the notion of domesticity. However, the scientific definition of domestication is an expression of the domestication syndrome [12,13]. Traditionally, human auto-domestication has usually been related to the changes in human behavior and lifestyles during the 'Neolithic revolution', with the introduction of agriculture and greater sedentariness [14–16]. Thus, the correlation of domestication with domos and the domicile pre-empted a scientific approach to the general issue until recently. Another limiting factor since Darwin [17] has been the implication that, typically, the domesticator has been the human species. This anthropocentrism is severely contradicted by the hundreds of other animal species, ranging from mammals to insects, that have domesticated other animal, plant, or fungi species. Moreover, domestication is a complex process involving symbiosis or mutualism [18–21] in many cases and can even involve aspects of gene–culture coevolution [22,23].

In vertebrate species, the domestication syndrome [12] is expressed by several universal features [24]. These include a reduction of tooth sizes and changes in craniofacial morphology, such as a shortened muzzle—or, in the case of humans, loss of prognathism. Others are alterations to ear and tail forms, shortening of the spine, reductions in total brain volume and specific brain regions, and depigmentation. Then there are alterations to adrenocorticotropic hormone levels and in concentrations of several neurotransmitters, sometimes accompanied by increased docility and tameness. Of distinctive consequences are the estrus cycles that occur more frequently or are non-seasonal and may even be eliminated entirely; and the preservation of a whole suite of typically neotenous effects, including juvenile behavior. Although it has been argued, based on experiences with foxes, that the concept of a domestication syndrome is inconclusive [25], others responded that the "family resemblance" among domesticates renders the notion useful [26,27]. However, they also proposed that rather than the domestication syndrome, emotional control and social motivation account for the changes in humans. More recently, it has been argued that the domestication syndrome is explained by shared reproductive disruption [28].

The domestication syndrome is facilitated by the mechanism of pleiotropy, an essential factor in domestication that defines when consistent selection for one gene affects two or more apparently unrelated traits in a population. For instance, when humans were the domesticators, they often selected in favor of lower flight response (docility), which introduced several other phenotypic traits coincidentally, such as facial architecture or reduction in dentition size [29].

## 2. Auto-Domestication of Hominins

The physiological changes from robust to gracile humans match those of vertebrate domestication, e.g., atrophy of brain volume and specific regions, dental reduction, the decline of prognathism, abolition of estrus, and probably depigmentation. Of particular significance is the progressive selection in favor of infantile physiology (neoteny or pedomorphosis, the attainment of sexual maturity before full somatic development), which became a dominant factor beginning around 40,000 years ago [10,11,30]. While it is apparent particularly in gracile humans, in numerous physiological and behavioral aspects, it needs to be emphasized that facial gracilization in humans has been evident for previous hundreds of millennia [31]. However, a distinctive rate of change occurred in the last third of the Late Pleistocene, in a relatively short time. It included a significant reduction in skeletal robusticity, especially in the cranium, smaller body size, more delicate skin, smaller mastoid features, flattened and broadened face, significantly reduced or absent tori, relatively large eyes, smallish nose, small teeth, a prolonged development period, less hair but retention of fetal hair, faster heartbeat, lower amount of energy expended at rest, increased longevity, higher pitch of voice, more forward tilt of head, more backward tilt of the pelvis, limbs that are proportionally short relative to the torso, narrower joints, and smoother ligament attachments [32]. We resemble the fetal chimpanzee more closely than

any other animal. For instance, most of our males lack a penis bone, as does the unborn chimpanzee, while the hymen of our females is retained for life (unless penetrated) but is a neonate feature in the ape. Similarly, in chimpanzees, the labia majora are an infantile feature, but in the human female are retained for life. In all apes, the lower abdomen's organs are aligned with the spines; but in humans and fetal apes, they point forward [33]. The unborn chimpanzee features a face almost as flat as a human's, whereas the adult ape shows distinctive prognathism. Human hands and feet differ significantly from the hands and feet of mature chimpanzees but resemble those of embryonic apes closely. While in chimpanzees, the arms become much longer after birth, this change does not occur in humans.

Humans' explorative behavior is probably a development derived from 'playful' conduct, as often observed in other domesticated mammals. In the human animal, it has prompted the development of paleoart production over the course of the second half of the Pleistocene and the facilitation of other improved cognitive performance. In particular, the introduction of iconicity in paleoart creation was prompted by juveniles [34,35], and the elaboration of exograms [36] generally served the proliferation of cultural complexity in the Upper Paleolithic. However, the greater part of the changes brought about by domestication have been detrimental to our lineage. This is an unwelcome observation, contradicting the self-aggrandizing tenor in much of archaeology that presents the human ascent as a teleology resulting in a 'crown of evolution'. Nevertheless, the empirical evidence is persuasive.

Most of the changes from robust to gracile hominins turned the concept of natural selection on its head. They offer no evolutionary benefits but mostly disadvantages—and mass migrations did not cause them, nor changing climate, sea level adjustments, language introduction, new technology, larger brains, cognitive revolution, significant dietary changes, fire use, population replacement, or any of the other reasons usually cited. The developments that led to graciles began about 40,000 years ago when hominin fossils reveal a distinctive trend towards physiological gracility. At that same time, a pronounced interest in female fecundity or sexuality began to be reflected in various forms of paleoart, such as figurines, pictograms, and petroglyphs of the Early Upper Paleolithic (Figure 1). A process of auto-domestication yielded less robust humans in every respect, who were subject to thousands of neuropathologies and various neurodegenerative disorders [11,30], to mental illnesses, brain atrophy, and consistent neotenization. Thus, the evidence suggests that the change from robust to gracile humans towards the end of the Pleistocene did not serve the teleology of human ascendancy but marks a significant deterioration of the human genome. For that reason alone, if for no other, it cannot result from evolution by natural selection. It was not a Darwinian process but one of Mendelian imperatives.

The auto-domestication hypothesis, which replaces the partially refuted replacement hypothesis, proposes that cultural practices had become so influential around forty millennia ago that factors attributable to learned behavior influenced mate choice. Mating imperatives are culturally governed, undeniably dominating sexual choice in present humans [37]. They must therefore have been introduced at some point in the past. Similarly, the modern human is the only animal with distinctive cultural mating preferences. These may involve not only personality traits but also anatomical qualities, including such cultural concepts as 'attractiveness' [38]. Again, it is self-evident that this must have been initiated at some point in our history. These preferences in mate selection are firmly ingrained in all known extant societies [39–46]. Another factor of interest is that our subspecies is the only mammal whose female attractiveness is often more important than male [39] and whose males appear to select mates. However, the mate selection strategies of humans are exceedingly intricate [47], and those of the distant past can only be conjectured.

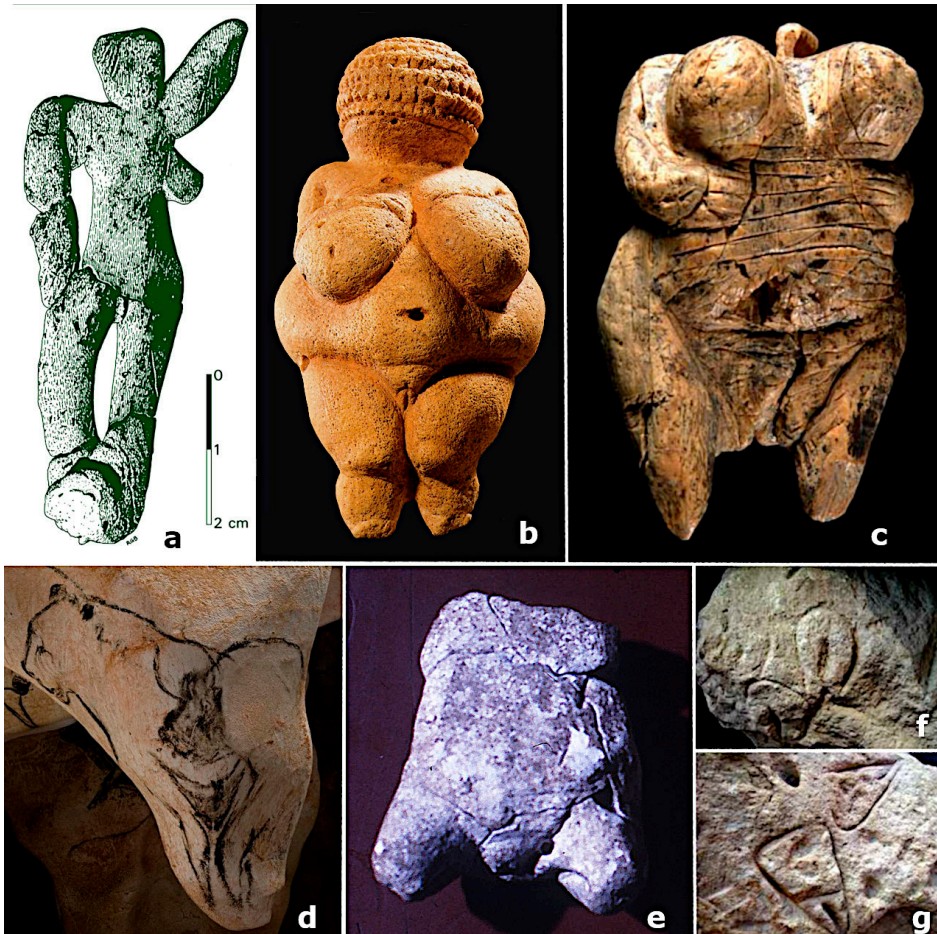

**Figure 1.** Some depictions of females or their genitalia of the Aurignacian and Gravettian, from (**a**) Galgenberg, (**b**) Willendorf, (**c**) Hohle Fels, (**d**) Chauvet Cave, (**e**) Avdeevo, (**f**) Abri Cellier, and (**g**) La Ferrassie. Reprinted from [30], © Robert G. Bednarik.

Consistent selection of females based on cultural constructs of attractiveness, i.e., their neotenous appearance, has two effects. It will progressively select in favor of female neoteny, and such selective breeding will introduce domestication, at least partially replacing natural selection. The genes of robust humans were not replaced by an intrusive population from Africa but by individuals considered attractive who had more offspring. This trend is evident in all four continents occupied by hominins at that time. The fossil record confirms that female humans led the transition from robust to gracile types between 40,000 and 30,000 years ago [10,11,30]; males lagged many millennia behind females (Figure 2).

Two other relevant human characteristics are the loss of estrus in the female and the introduction of menopause. The latter is almost unique to humans, shared only with four cetaceans and possibly some primates [48]. Female fecundity ends with menopause, which may explain why Paleolithic societies would have attached importance to youthful female appearance deriving from neoteny [43]. While most of the universal preferences observed in the mate choice of modern humans seem to serve no processes of natural selection, youth does, in offering better procreative potential through more prolonged remaining fertility. Therefore, the illusion of youth inherent in neotenous characteristics may have contributed to the neotenization of final Pleistocene hominins and the effect of incidental auto-domestication. Similarly, the loss of estrus is probably a phenomenon of the domestication syndrome but could have a more extended history. Females who were receptive for longer periods would be expected to conceive more frequently than others and would be more likely to have been provisioned with protein and fat needed for successful and frequent pregnancies [49–54]. This does not contradict the "grandmother

hypothesis" [55], which endeavors to explain the large fraction of post-fertile years women live [56,57].

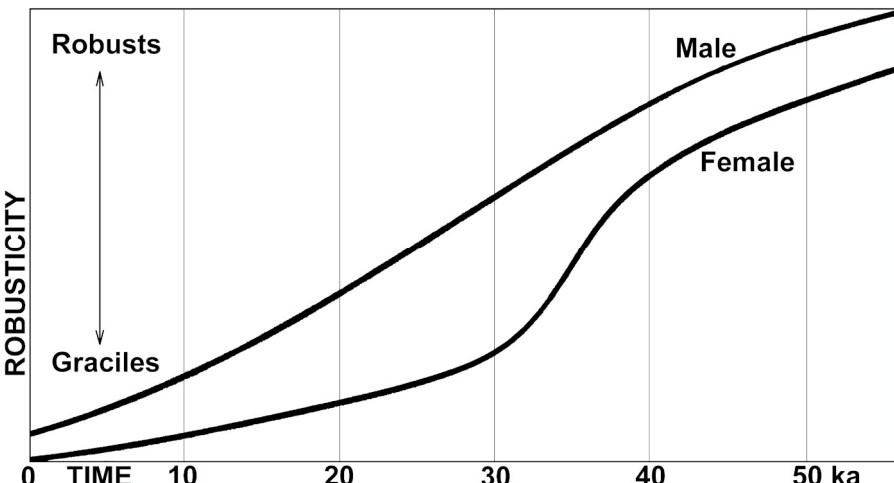

**Figure 2.** Male and female relative cranial robusticity/gracility in Europe during the final Pleistocene. Reprinted from [30], © Robert G. Bednarik.

### 3. Testing the Domestication Hypothesis

Testable hypotheses are assessed by subjecting them to falsification attempts. The human self-domestication theory can be checked by various strategies. For instance, one can establish the genetic markers suggesting domestication and then attempt their detection in the genomes of extant humans while verifying their absence in pre-domestication robust humans. When the domestication theory was first formalized in 2008, analyses of the human genome had begun, but no genetic markers of domestication were known. In the subsequent years, some data of this nature have gradually become available. It implicitly validates the auto-domestication hypothesis, which has garnered some supporters in recent years. Having been reviewed [58,59] and corroborated by others [60–62], it has emerged as a viable replacement of the replacement hypothesis. This is not surprising: the latter was based on a hoax by an archaeology professor in the 1970s [63,64], replete with fake datings of numerous human skeletal remains [11,65]. The refutation of one of its essential postulates that robusts and graciles were not interfertile has challenged it severely, and the domestication hypothesis has rendered it obsolete.

Dozens of overlapping genes have recently been revealed by selective sweeps in the genomes of modern humans and several domesticated mammalian species [4,66,67]. For instance, the domesticated horse shares seven genes with extant humans, cattle and humans share nine genes, and the cat and the dog share fifteen genes with us. Forty-one genes under positive selection have been identified, in extant humans and in one or more of the four domesticates that were considered in the study. That does not necessarily prove that domestication proceeded analogously in the five species. The circumstances of domestication can be assumed to have differed in each species affected by it. Nevertheless, the genes that have been established to be shared by domesticate animals and *Homo sapiens sapiens* suggest that the latter was subjected to changes resembling those experienced by mammalian domesticates. However, caution demands one to acknowledge that only some of these genes are shared across all five domesticates, while numerous genes are under selection in various domesticates but perhaps not in humans.

Of relevance is also that none of the 17,367 protein-coding genes found in the remains of two Neanderthals [68] is listed among those known to overlap between at least two domesticated species (*SMG6, PLAC8L1, ADAMTS13, BRAF, RNPC3, SEC24A, CLEC5A, FAM172A, VEZT, NRG2, GRIK3, STK10, ATXN7L1, DCC,* and *TMEM132D*). The pre-domestication status of *Homo sapiens neanderthalensis* appears to be confirmed by this finding. An earlier study sequencing the mitochondrial DNA genomes of dogs, wolves,

and coyotes [69] had already found that nonsynonymous changes in mitochondrial genes have accumulated at a faster rate in dogs than in wolves. This seems to explain the extreme phenotypic diversity among dogs. Moreover, deleterious mutations have accumulated as a legacy of domestication in the form of a higher proportion of nonsynonymous alleles than non-functional genetic variation [70]. Like population bottlenecks and inbreeding, artificial selection increases deleterious genetic variation levels [71] rather than yield a superior new species as postulated by the replacement hypothesis for the first two variables. The increased burden of deleterious variants deriving from domestication is estimated to be 2–3% higher in dogs than grey wolves. A recent study compares whole-genome resequencing data from dogs, pigs, rabbits, chickens, silkworms, rice, and soybean with their wild progenitors [72]. It reports much lower genetic variation across a range of allele frequencies, but nonsynonymous amino acid changes are increased in all except one domestic species.

An alternative genetic approach to test the human auto-domestication hypothesis is to detect detrimental genes deriving from domestication in robust humans. Examples are the genes *AUTS2* and *CADPS2*, involved in autism, and *NRG1* and *NRG3* (both schizophrenia), which were all not reported in 'Neanderthals' [73]. The microcephalin D allele was only established in the final Pleistocene, so it cannot have been present in robust humans [74]. Another contributor to microcephaly, the *ASPM* allele, only appeared in the mid-Holocene [75]. Genes *RUNX2* and *CBRA1*, responsible for dental abnormalities and malformed clavicles, or the mutation *THADA*, causing type 2 diabetes, are also unlikely to predate human self-domestication. Indeed, natural selection cannot account for the many thousands of genetic disorders of modern humans (for instance, as of 2019, the molecular basis was known of 6328 Mendelian disorder phenotypes alone and 4017 genes with phenotype-causing mutation). Moreover, contrary to some views, other extant primates lack genetically based mental or neurodegenerative illnesses, so it is assumed that these were introduced at some point in hominin history. Natural selection would probably have selected against mutations disadvantaging their hosts severely [11,30]. Therefore, the accumulation of thousands of detrimental traits in extant humans suggests a suspension of Darwinian principles.

A crucial relevant development has been the discovery that the chromatin remodeler BAZ1B in neural stem cells controls the evolution of the modern human face [76]. Contrary to [28] and other alternatives proposed, neural stem cells are deeply involved in domestication [13]. The human face is the result of mild neurocristopathy, and the modelling of paleogenomic and in vitro sampling (Williams–Beuren Syndrome) provided a coherent explanation of the genetic mechanisms of domestication. Thus, the 7q11.23 region is relevant not only to neurodevelopment disease modelling but also to domestication genetics, where it confirms that human self-domestication did occur.

## 4. Conclusions

The human auto-domestication hypothesis postulates that its defining process began with relatively minor behavioral modifications. Towards the last quarter of the Late Pleistocene, human mating preferences previously unheard of in the animal kingdom developed gradually. Males acquired a preference for females of neotenous characteristics, perhaps as a response to menopause. In this, they overturned the hitherto universal principle that it is the females who, directly or indirectly, select mating partners. The consistent selective breeding introduced the domestication syndrome, self-reinforcing the trend in a feedback loop. This changed the human genome decisively and irreversibly within such a relatively short time (in the order of twenty millennia) that paleoanthropologists perceived the sudden appearance of a new species that must have come from 'elsewhere'. Once the syndrome had been established, it replaced Darwinian natural selection with Mendelian principles as the primary agent of genetic changes. It introduced numerous, mostly deleterious traits that typify the human condition as we know it [11].

The significant neotenization defining extant humans results from the domestication syndrome's pleiotropy [30]. Contrary to the views of some, domestication is indeed not

defined by who does what to whom. In a scientific sense, it merely refers to mutations expressing traits of the domestication syndrome. The roles of the constellation of neoteny, estrus, menopause, and youth remain open to further possible explanations. However, the underlying principle that the domestication syndrome accounts for the change from robust to gracile hominins is proposed to have been soundly established.

The renewal this realization introduces in the field of human evolution may also facilitate enlivening new insights for that discipline. For instance, it has been consistently postulated that depigmentation is a response to the migration to northern latitudes and climates. Perhaps it is, but it could just as easily be a domestication reaction that was omitted as unfavorable in the low latitudes. Similarly, instead of our foot being an adaptation to upright walking, our erect locomotion could have been facilitated by the architecture of the neotenous foot. Indications of hominin neoteny can be traced back as far as Ardipithecus [77]. The human auto-domestication hypothesis also provides the academic cottage industry of predicting the human future with ample opportunities to develop new strands of reasoning, heralding as it does the accelerating and irreversible decline of the human genome.

**Funding:** This research received no external funding.

**Institutional Review Board Statement:** Not applicable.

**Informed Consent Statement:** Not applicable.

**Data Availability Statement:** Not applicable.

**Conflicts of Interest:** The author declares no conflict of interest.

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
