# Peer review of "The Domestication of Humans"

_encyclopedia, doi:10.3390/encyclopedia3030067_

Round 1
Reviewer 1 Report
Dear Editor of Encyclopedia and Author, I believe that the entry is of good quality and is ready for publication.
My best regards.
Author Response
Reviewer 1 thinks the paper is of good quality and ready for publication.
Reviewer 2 Report
This article presents the human autodomestication hypothesis, suggesting that human mating preferences led to behavioral modifications and a domestication syndrome that changed the human genome. I have no major concerns and the article is already well-written and well supported by references. There are a few comments to improve the paper.
Gene names (pages 5-6) and species names (Homo sapiens sapiens, Homo sapiens neanderthalensis, etc.) must be italicized throughout the paper.
When explaining the domestication syndrome (lines 64-67), it would be appropriate to briefly explain why these changes occur. Also explain the selection pressures that favor these traits.
No concern.
Author Response
Gene names and species names have been italicized. Concerning the recommendation to explain why the domestication syndrome-generated changes occurred: some clues are in the text, but our genetic understanding of this is far too limited to go into such details in an encyclopedia entry.
Reviewer 3 Report
This article is composed of four main sections:1. Introduction, 2. Auto-domestication of hominins, 3. Testing of domestication hypothesis, 4. Conclusion.
Each section contains several evidences to support the title of the section.
However, I have one major and three minor comments to this manuscript.
Major comment is the contents of each section does not seem to have a message to readers. It is not clear that what the author want to deliver.
The followings are minor comments.
1) I do not agree with the description of line 79-84. The author seems to confuse pleiotropic effects with polygenic effects. At least, a cause of Marfan syndrome seem to be polygenic. It is naturally expected those causal genes have pleiotropic effects. But the Marfan syndrome is a polygenic disease.
2) Line 209-245, it is fine that the specific gene names were listed. But it is not clear to readers, function of these genes. It is better at least to show example of function for a gene and explain what is the contribution of the gene to domestication. And it is unlikely sharing the mutant origin between humans and domesticated animals.
3) After reading this manuscript I had a vague impression that the author considered that detrimental mutations had been accumulated in the modern human populations and finally this process made our genomes decline. The author seems to consider that this is the effect of "domestication". But the flow of the argument seems to be subjective. I recommend to argue this not qualitatively but quantitatively.
Author Response
This is not a research paper but an encyclopedia entry. It presents the relevant empirical information, arguments and counter-arguments and has a chapter on Conclusions.
Comment 1: It is near-impossible to confuse pleiotropic with polygenic effects, but the two are interwoven in their expressions. Pleiotropy describes when one gene causes apparently unrelated multiple phenotypic expressions, as is the case in domestication: selection for one attribute can cause a cascade of physical characteristics to change. I have modified the text to avoid confusion.
Comment 2: The reviewer misunderstands the purpose of listing these genes, which is, as stated, to list genes attributed to the domestication syndrome, not to show their individual functions, which we understand inadequately.
Comment 3: This “vague impression” is correct: most of the mutations arising from the domestication syndrome are considered detrimental to the genome of the domesticates. And there is no need to argue about the issue: some of the references cited have already established that humans experienced domestication. This is merely an encyclopedia entry, not a debate article.
Reviewer 4 Report
"The manuscript titled 'The Domestication of Humans' addresses a captivating topic: the auto-domestication of Anatomically Modern Humans (AMH). However, this manuscript overlooks a significant amount of recent literature. For instance, studies by Cieri et al. (2014), Hare (2017), and Bruner and Gleeson (2019) are not taken into account. Furthermore, the concept of self-domestication in humans is contested by some scholars, as the social evolution of humans can also be explained by selection for pro-social motivation and self-control, which are influenced by symbolic communication and representation (Shilton et al., 2020).
I have several major concerns with this text:
1) Auto-domestication and the gracilization of AMH started only 40,000 years ago (lines 88-91).
However, facial gracilization began much earlier, at least 315,000 years ago, as detailed analyses of facial and cranial human remains from Irhoud (Morocco) have shown. The facial morphology of the Irhoud fossils is nearly indistinguishable from that of recent Modern Humans, and the Irhoud fossils represent the early phase of H. sapiens evolution in Africa (Hublin et al., 2017). Lacruz et al. (2019) also indicate that the facial characteristics of H. antecessor, a key Last Common Ancestor (LCA) candidate of AMH and Neanderthals, exhibit modern features while retaining some primitive characters in other parts of the cranium. I strongly advise the author to include the ancient origin of the modern human face in this manuscript.
2) The manuscript simplifies the explanation of the domestication syndrome by stating that it is acquired through the mechanism of pleiotropy (line 74).
This is an oversimplified explanation. According to Wilkins et al. (2021), the domestication syndrome refers to a set of unexpected physical differences that frequently appear in different domesticated mammals. Gleeson and Wilson (2023) argue that shared domestication syndrome traits do not necessarily require shared genetic mechanisms or pleiotropy. Instead, they propose that changes in wild selective regimes, such as disrupted sexual selection, changes in resource availability and predation pressure, and intensified maternal stress, can cause these traits. These altered selective influences can include a selection for tameness. It would be interesting if the author could investigate whether such selective shifts could have occurred in the evolution of AMH.
3) The origin of menopause (lines 168-180).
The author attributes the origin of menopause to the fact that it is costly to feed pregnant females, stating, "Females who were receptive for longer periods would be expected to conceive more frequently than others and would be more likely to have been provisioned with protein and fat needed for successful and frequent pregnancies" (lines 177-179). I suggest that the author also consider the grandmother theory, which explains menopause and the post-fertility years in women. Hawkes (2020) provides an excellent review on this topic. In all extant hominids, female fertility ends at approximately the same age (around 45 years), suggesting that the onset of menopause is likely an ancestral feature. However, women in various human populations often live long after their fertility ends. Significantly, a large portion of human female-adult-years is lived after menopause, and post-fertile women contribute to the reproduction of their genes through grandmothering (Hawkes, 2020).
4) The dismissal of female agency in mate choice.
The statement in lines 151-154 that our subspecies is the only mammal in which female attractiveness is universally more important than male and males select mates rather than vice versa is somewhat disappointing. Recent studies have highlighted that women prefer to choose partners who display cues of fitness (Frederick et al., 2013). Moreover, Frederick et al. (2013) and Buss and Schmitt (2019) extensively discuss female agency in mate choice. It would be beneficial for the author to reference these papers or similar ones.
Suggested references:
- Bruner E, Gleeson BT. 2019. Body cognition and self-domestication in human evolution. Front. Psychol. 10, 1111.
- Buss, D.M., Schmitt, D.P. 2019. Mate Preferences and Their Behavioral Manifestations. Annual Review of Psychology 70, 23.1–23.34.
- Cieri RL, Churchill SE, Franciscus RG, Tan J, Hare B. 2014. Craniofacial feminization, social tolerance, and the origins of behavioral modernity. Curr. Anthropol. 55, 419–443.
- Frederick, D., Reynolds, T., Fisher, M. 2013. The importance of female choice: Evolutionary perspectives on constraints, expressions, and variations. In M. L. Fisher, J. R. Garcia, and R. S. Chang (Eds.), Evolution's empress: Darwinian perspectives on the nature of women (pp. 304-329). New York, NY: Oxford University Press.
- Gleeson BT, Wilson LAB. 2023. Shared reproductive disruption, not neural crest or tameness, explains the domestication syndrome. Proc. R. Soc. B 290: 20222464. https://doi.org/10.1098/rspb.2022.2464.
- Hare B. 2017. Survival of the friendliest: Homo sapiens evolved via selection for prosociality. Annu. Rev. Psychol. 68, 155–186.
- Hawkes K. 2020. The centrality of ancestral grandmothering in human evolution. Integr. Comp. Biol. 60, 765-781.
- Hublin, J.J., Ben-Ncer, A., Bailey, S.E., Freidline, S.E., Neubauer, S., Skinner, M.M., Bergmann, I., Le Cabec, A., Benazzi, S., Harvati, K., et al. 2017. New fossils from Jebel Irhoud, Morocco and the Pan-African origin of Homo sapiens. Nature 546, 289–292.
- Lacruz RS, Stringer CS, Kimbel WH, et al. 2019. The evolutionary history of the human face. Nature Ecology & Evolution 3, 726–736.
- Shilton D., Breski M., Dor D., Jablonka E. 2020. Human Social Evolution: Self-Domestication or Self-Control? Front. Psychol. 11:134.
- Wilkins AS, Wrangham R, Fitch WT. 2021. The neural crest/domestication syndrome hypothesis, explained: reply to Johnsson, Henriksen, and Wright. Genetics 219, 7."
Author Response
Nine new references have been introduced, eight of them recent. For instance, Bruner and Gleeson (2019) is shown to have been refuted effectively, the role of neural crest cells has been reinforced by genetics work, and the domestication theory has been further validated.
Comment 1: The issue has been clarified, and more detail has been added concerning the genetic origin of the human face.
Comment 2: Wilkins et al. are now cited twice and approvingly. It is also explained why Gleeson and Wilson are probably wrong. The pleiotropy passage has been modified.
Comment 3: Agreed and adopted, but it has no real effect on the domestication issue.
Comment 4: The text has been revised in accordance with the recommendation.
Round 2
Reviewer 2 Report
Thank you for the revised version. I have no more concerns. Congratulations!
Reviewer 3 Report
Thank you very much for your reply to my review.
I realized my misunderstand of an article in "Encyclopedia", sorry for this misunderstanding.
Now I realized that your article was appropriate to be published in "Encyclopedia".
Reviewer 4 Report
The author has integrated most of the suggested references.